# 3D printable diffractive optical elements by liquid immersion

Reut Orange-Kedem[1,2], Elias Nehme[2,3], Lucien E. Weiss [2,4], Boris Ferdman[1,2], Onit Alalouf[2,4], Nadav Opatovski[1,2] & Yoav Shechtman [1,2,4✉]

Diffractive optical elements (DOEs) are used to shape the wavefront of incident light. This can be used to generate practically any pattern of interest, albeit with varying efficiency. A fundamental challenge associated with DOEs comes from the nanoscale-precision requirements for their fabrication. Here we demonstrate a method to controllably scale up the relevant feature dimensions of a device from tens-of-nanometers to tens-of-microns by immersing the DOEs in a near-index-matched solution. This makes it possible to utilize modern 3D-printing technologies for fabrication, thereby significantly simplifying the production of DOEs and decreasing costs by orders of magnitude, without hindering performance. We demonstrate the tunability of our design for varying experimental conditions, and the suitability of this approach to ultrasensitive applications by localizing the 3D positions of single molecules in cells using our microscale fabricated optical element to modify the point-spread-function (PSF) of a microscope.

[1] Russell Berrie Nanotechnology Institute, Technion—Israel Institute of Technology, Haifa, Israel. [2] Lorry Lokey Interdisciplinary Center for Life Sciences and Engineering, Technion—Israel Institute of Technology, Haifa, Israel. [3] Department of Electrical Engineering, Technion—Israel Institute of Technology, Haifa, Israel. [4] Department of Biomedical Engineering, Technion—Israel Institute of Technology, Haifa, Israel. ✉email: yoavsh@bm.technion.ac.il

DOEs are ubiquitous in optics, serving a variety of purposes in wavefront shaping[1], including laser processing[2], lithography, and holographic lighting[3,4], photonic structures[5], solar power[6], communication[7], and sensitive microscopy[8–11]. In localization microscopy, where images of nanoscale fluorescent emitters are processed to recover precise spatial information, DOEs are especially useful for modifying the imaging system's point-spread function (PSF) in order to efficiently encode additional physical properties, such as the depth[8,12,13] color[14], and orientation[15]. This is typically accomplished by expanding the imaging path of the microscope with two additional lenses and adding a phase mask in between at the conjugate back focal plane (Fig. 1a). The function of this mask is to precisely adjust the light's accumulated phase by controlling the relative path length at different positions that, at the back focal plane, correspond to different spatial frequencies. To accomplish this, two classes of devices exist, (1) controllable optics, such as liquid crystal spatial light modulators[16] (LC-SLMs) and deformable mirrors;[17] (2) prefabricated dielectric masks[18], which contain feature sizes λ/10, i.e., tens-of-nanometers for visible light. Both approaches have drawbacks: LC-SLMs modulate only one polarization, which in fluorescence microscopy translates to low photon efficiency (<50%); dielectric masks require a cumbersome nano-fabrication process due to the stringent fabrication constraints. In addition, both approaches are expensive, costing thousands to tens of thousands of dollars per device.

Combining optical materials with different refractive indices has been shown to be useful in a variety of applications, from aberration correction in lenses[19], through Bragg grating fabrication optimization[20] and simple fabrication of solid DOEs[21], to high numerical aperture microscopy[22].

Here, we demonstrate the production of high-quality, modifiable DOEs with flexible designs and feature sizes orders of magnitude larger than the traditional nanoscale, thereby drastically simplifying production and lowering manufacturing costs by orders of magnitude while enabling tunable DOE modification. This is accomplished by immersing a dielectric element in a nearly-index matched liquid medium, such that accumulating the desired phase-delay difference incurred by passing through the device requires propagating through proportionally more material (Fig. 1c, d).

## Results

**Concept and fabrication of liquid immersed DOE.** The relation between optical phase accumulated to element thickness can be described as follows:

$$\Delta\phi = \frac{2\pi h}{\lambda}\left(n_{\mathrm{DOE}} - n_{\mathrm{media}}\right) \tag{1}$$

Where $\Delta\phi$ is the accumulated phase difference between light traversing the DOE compared to the surrounding media; $\lambda$ is the wavelength; $h$ is the height of the DOE; and $n$ refers to the

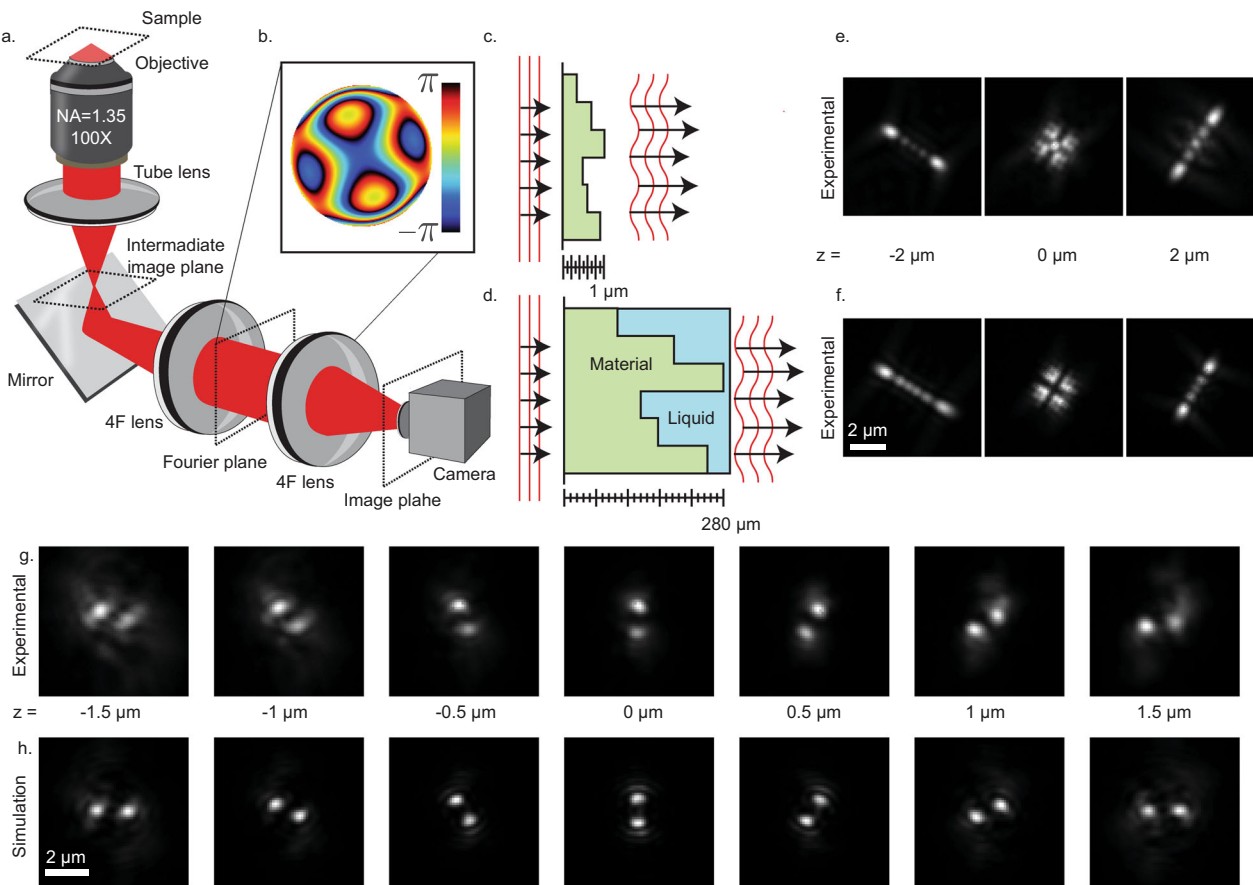

**Fig. 1 The concept of the liquid immersed phase mask. a** The optical system, an inverted microscope expanded with a 4F system. **b** A phase mask, implemented in the Fourier plane. **c** Schematic of a conventional photolithographically fabricated phase mask (side view), with etching depth up to 1 μm. **d** Schematic of a liquid immersed phase mask (side view), fabricated with additive manufacturing, with height up to 280 μm. **e, f** Experimental images of a fluorescent microsphere at different axial positions, using the two different phase masks. Note that the small differences in the PSF shape are derived from the differences in mask designs (Supplementary Fig. 1). **g** Experimental and (**h**) simulated images of a fluorescent microsphere at different axial positions, using a liquid-immersed Double Helix phase mask.

refractive index of the DOE and surrounding media, respectively. Normally, the DOE is made of high-quality optical material, e.g., Fused silica, and $n_{DOE} \cong 1.45$; critically, normally, the surrounding media is air, $n_{media} \cong 1$. On the other hand, in our immersed device, the air is replaced by a liquid, which decreases the refractive index difference by orders of magnitude to be $n_{DOE} - n_{media} \approx 2 \cdot 10^{-3}$. This means that if the mask is immersed in liquid, $h$ needs to increase significantly to maintain the same phase accumulation as in the air. In other words, the axial size of the DOE scales up by orders of magnitude.

Notably, the key feature that enables simple fabrication of the scaled-up DOE is the following: the decrease in refractive index difference not only increases the required height of the DOE—importantly, the tolerable height error, i.e., fabrication error, scales similarly. The effects of the index and height errors on the accumulated phase error can be described by:

$$
\begin{aligned}
\delta\Delta\phi_{Error} &= \Delta\phi_{actual} - \Delta\phi_{target} \\
&= \frac{2\pi}{\lambda}(\Delta n + \delta n) \cdot (h + \delta h) - \frac{2\pi}{\lambda} \\
&\cdot \Delta n \cdot h \xrightarrow{\delta n, \delta h \ll \Delta n, h} \frac{2\pi}{\lambda} \cdot \Delta n \cdot \delta h + \frac{2\pi}{\lambda} \cdot h \cdot \delta n
\end{aligned}
\tag{2}
$$

Where $\delta\Delta\phi_{Error}$ is the unintended accumulated phase difference between light traversing the DOE compared to the surrounding media; $\lambda$ is the wavelength; $\delta h$ is the height error from the fabrication of the DOE; and $\delta n$ refers to the error in refractive index difference between the DOE and surrounding media. From the relation in Eq. (2), we can conclude that as the refractive index difference decreases, the effect of the fabrication error is similarly reduced. This enables low precision 3D printing fabrication methods, and guarantees similar relative phase error as conventional lithography dielectric phase mask.

To fabricate the phase mask, we printed a ceramic mold with an *XYZ* resolution of tens of microns (Supplementary Note 4). Next, we cast a transparent polymer with a refractive index of ~1.45. This polymer DOE was bonded to a fused silica wafer, which was then placed in a chamber and filled with immersion liquid (a glycerol-water mixture). This assembly was then aligned in the back focal plane of a fluorescence microscope, and further optimized by changing the immersion media to achieve the desired properties (Supplementary Fig. 2).

To demonstrate broad versatility in possible mask design, two fundamentally different patterns were fabricated. First, we fabricated the Tetrapod phase mask, which consists of a relatively smooth surface. Second, we fabricated and deployed the Double Helix mask, containing sharp discontinuities and non-smooth regions (Supplementary Fig. 1). The experimental PSFs were measured by axially scanning fluorescent microspheres fixed to the surface (Fig. 1f, g), and comparing their performance to a conventional photolithographically fabricated Tetrapod phase mask (Fig. 1e) and a computationally generated image stack for the Double helix mask (Fig. 1h). These fabricated masks demonstrate that smooth designs and also designs with sharp edges and discontinuities can be implemented using our method with simple adjustments.

**Tunability of liquid immersed phase mask**. One key advantage of immersing a DOE in a liquid is that the liquid properties are easily controllable, which allows tunability. Here, we consider the Tetrapod phase-mask, which has been developed to optimally encode the 3D position of point emitters over a large and dynamic range[13,23]. The useful depth range of the Tetrapod PSF scales with the height of the phase-delay pattern, where a shorter range allows higher localization precision. A controllable device, such as a deformable mirror of an LC-SLM, can be modified to

match the requirements of a specific experiment by changing the amplitude of the mask;[23] however, for fabricated devices, the precise properties must be chosen beforehand[18,24]. For our liquid-immersion device, however, the relative phase delay can be changed by simply replacing the immersion media. To demonstrate this tunability, we deploy our 3D-printed device to the problem of 3D single-particle tracking.

In brief, a 3D imaging chamber containing fluorescent nanospheres was created by sandwiching a droplet between two glass slides with a double-sided adhesive spacer (Fig. 2b, c). For the standard PSF, where the depth range is several hundred nanometers, the freely diffusing beads move quickly out of focus. With the Tetrapod phase mask, these particles can be tracked much longer and in 3D. By changing the phase mask's immersion media, we can adjust the applicable depth range of the Tetrapod PSF (Fig. 2d, e). While a smaller range Tetrapod yields a smaller PSF footprint and higher spatial precision, a larger range PSF can be used to attain longer trajectories (Fig. 2f, g). Using this capability, the axial range of the mask can be adjusted to match a set of experimental conditions, e.g., the height of a specific cell.

**3D single-molecule localization microscopy (SMLM) using liquid immersed DOE**. For low-light applications the photon efficiency of the DOE is paramount. To compare to the leading alternatives, we first quantified the efficiency of our immersion-based microscale mask relative to a nanoscale photo-lithographically made mask, by measuring the intensities of laser light modulated by both masks, and found equivalent photon efficiencies (Supplementary Note 3).

Next, to illustrate that the high efficiency of our device makes it suitable even for the most photon-demanding ultrasensitive applications, we demonstrate super-resolution, SMLM in fluorescently labeled cells. Human osteosarcoma epithelial (U2OS) cells were cultured on glass coverslips, then chemically fixed and labeled with the mitochondria-binding antibody αTOMM20, or a microtubule-binding antibody pair that binds to α-tubulin, β-tubulin, both directly conjugated to Alexa Fluor 647 (see "Methods"). Cells were washed and placed in an imaging buffer that promotes stochastic blinking under high-intensity (13,000 kW/cm²) 640 nm light suitable for dSTORM[25]. Movies containing ~50,000 frames of blinking emitters were analyzed using the Deep-STORM3D[26] neural-net-based localization algorithm.

The range afforded by the Tetrapod mask was adjusted to enable the whole cell to be imaged simultaneously (Fig. 3a, b), while still maintaining the precision to visualize the key features in the biological structures (Fig. 3d, f, g). Figure 3d, e demonstrates qualitatively that our proposed design enables resolving mitochondrial hollow structures, in a similar fashion to results achieved with either a deformable mirror[17] or a diffractive optical element[26] implementation. Furthermore, Fig. 3f, g demonstrates a Gaussian fit to a single microtubule from our reconstruction which leads to a full width at half maximum (FWHM) of 84 nm laterally and 98 nm axially. The apparent width of a microtubule filament in SMLM depends both on the localization precision and on the diameter of the fluorophore distribution around the microtubule center[27]. Modeling microtubules as cylindrical structures with a diameter of ~25 nm, labeled with fluorophores attached via an antibody of size ~12.5 nm[27], indicate that our lateral and axial localization precisions are ~17 nm and ~24 nm respectively. This result is in line with previous work employing a deformable mirror implementation[17]. In addition, in Fig. 3h we quantify the lateral resolution of the microtubules reconstruction to be ~77 nm using a parameter-free image resolution estimation algorithm based on decorrelation analysis[28].

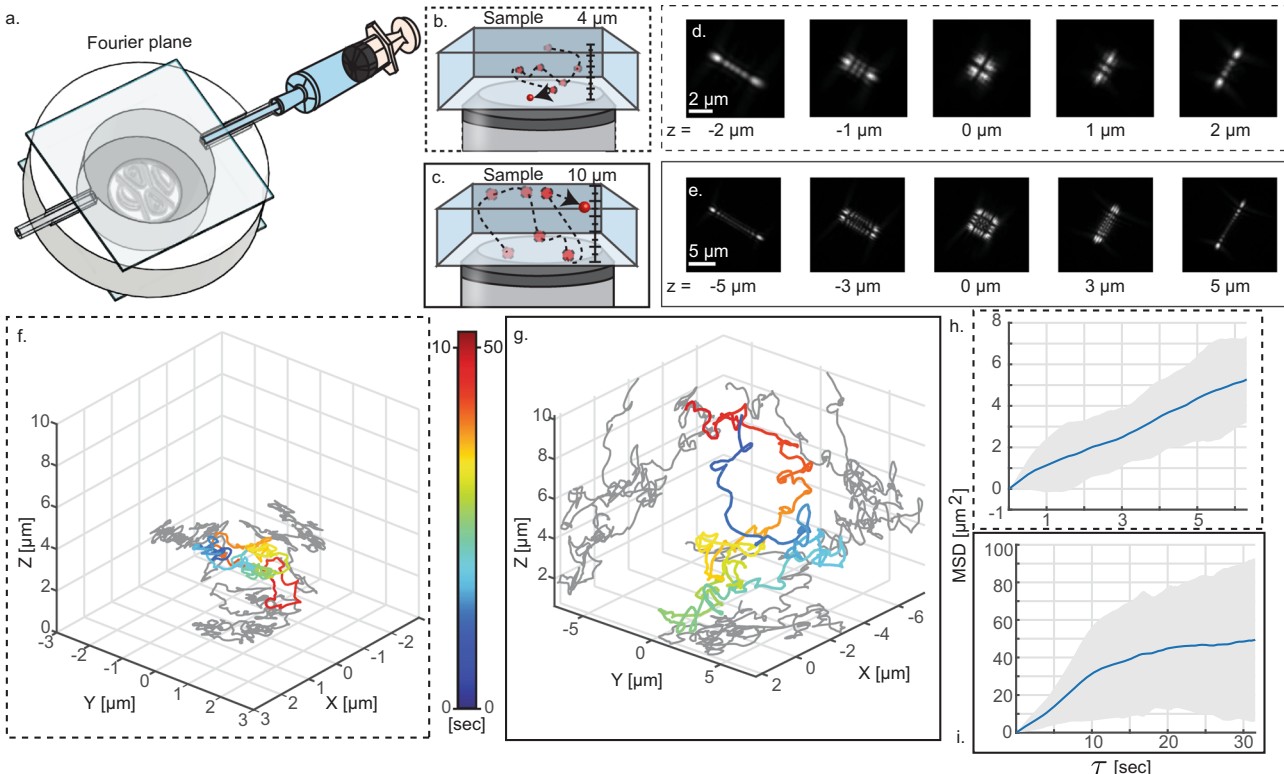

**Fig. 2 The results of 3D single-particle tracking experiments. a** Illustration of Z-range tunability by liquid replacement. **b, c** Different tracking ranges 4 and 10 μm, respectively, obtained using the same device with different liquids ($n_{ref} = 1.41560$, 1.41897). **d, e** Corresponding experimental Z-stacks of a fluorescent microsphere. **f, g** The reconstructed trajectories in different ranges respectively. A larger depth range enables longer observation. **h, i** Corresponding mean squared displacement (MSD) plots. Dashed (continuous) lines refer to the 4 μm (10 μm) range.

## Discussion

Here, we have demonstrated that liquid immersion can be used to transform the canonical design restrictions of fabricated DOEs by modifying the ratio between path length and accumulated phase differences. In doing so, we have upscaled the desired feature size from nano- to microscale, accessing the regime needed to utilize inexpensive 3D printing technology, which is becoming increasingly available. Furthermore, we have shown that this can be done at no cost in terms of the performance quality, making the approach broadly applicable to the many areas where diffractive optics are used, including range sensing-LIDAR/LADAR applications[29], optical information processing[30], Bragg grating fabrication[31], and more.

## Methods

**Optical setup.** The optical setup was built on an inverted fluorescence microscope (Eclipse Ti2, Nikon), where the emission path was expanded with a 4F system, that is two lenses, $f = 20$ cm, with a conjugate back focal plane of the objective in between, i.e., where the phase mask is placed. To illuminate the sample, a 640 nm laser (Obis, Coherent) was used in epi-illumination mode, though a 100X, NA 1.35, Silicon-oil objective (Lambda S 100XC Sil, Nikon). The emission light was collected back through the same objective and spectrally filtered with a bandpass filter (Quint-pass 405/445/514/561/640, Chroma). After exiting the microscope, the emission light was aligned through the fabricated phase mask placed on a translatable mount (5-Axis Kinematic Mount-K5X1, Thorlabs), before being imaged on an sCMOS detector (Prime 95b, Photometrics).

**3D printing and molding a phase mask.** A detailed description of the fabrication process is described in the Supplementary Information. Briefly, a ceramic mold of the reverse pattern of the phase mask was printed via additive manufacturing (Xjet Ltd. Israel). A polydimethylsiloxane (PDMS) cast was made and bonded to a high-quality fused silica optical glass (Siegert Wafer, Germany) via a brief plasma treatment to both the wafer and back of the PDMS mask.

To construct the liquid-immersion chamber, a PDMS frame was molded to a custom machined metal disk. To ensure tight-fitting with the inlet and outlet tubes, the frame was polymerized with a tube inserted in the disk, which was cut after polymerization. The frame was then attached to both silica wafers, again by brief plasma treatments. This assembly is then used as the liquid-immersion chamber that contains the mask. The final step is to attach a light-blocking aperture to the back of the silica wafer in order to reject light beyond the size of the mask. This was done with UV-cured optical adhesive (NOA68T, Norland Products). The mask is then ready to be added to the optical system.

**Adding and exchanging immersion liquid into the device.** To prepare the chamber for the injection of liquid, it was filled with isopropanol. This helps (1) reduce air bubbles and (2) remove any residual immersion liquid from previous uses. Next, the desired RI of the immersion liquid was prepared by mixing glycerol and water and injecting it into the chamber. At each step, the optical performance of the system was compared to the expected performance mainly by a comparison of the 3D-PSF range, i.e., the mask range is increased with a larger RI difference.

**3D single tracking particle experiment.** To track fluorescent beads in 3D, we constructed an imaging chamber, $h = 250$ μm, consisting of two glass coverslips (#1.5 ultrafine, Marienfeld) bonded with double-sided adhesive (Gene Frame, 25 μL/1 cm², ThermoFisher). Fluorescent beads (200 nm 625/645 Fluorospheres, Invitrogen) were diluted in 25 μL of a glycerol-water mixture and added to the chamber before sealing it. Images were acquired with a 10 or 50 ms exposure time for the smaller and larger Z-range experiments, respectively. To increase the z-range of the PSF during the experiment, the refractive index in the mask was increased, in order to effectively increase the height of the mask. The dynamic positions of the recorded beads were then localized in each frame using MLE with an imaging model derived using the VIPR phase-retrieval algorithm[32].

**Cell culture and labeling for STORM experiments.** To prepare cells for imaging, cover glasses (#1.5H, 22X22 mm, Marienfeld) were cleaned in an ultrasonic bath (DCG-120H, mrc) with 5% Decon90 at 60 °C for 30 min, then washed with water, incubated in ethanol absolute for 30 min and sterilized with 70% filtered ethanol for 30 min. The slides were then seeded with U2OS cells and grown for 24 h in a six-well plate using phenol-free Dulbecco's Modified Eagle's medium (Gibco) with 1 g/l D-Glucose (i.e., low-glucose conditions), supplemented with fetal bovine

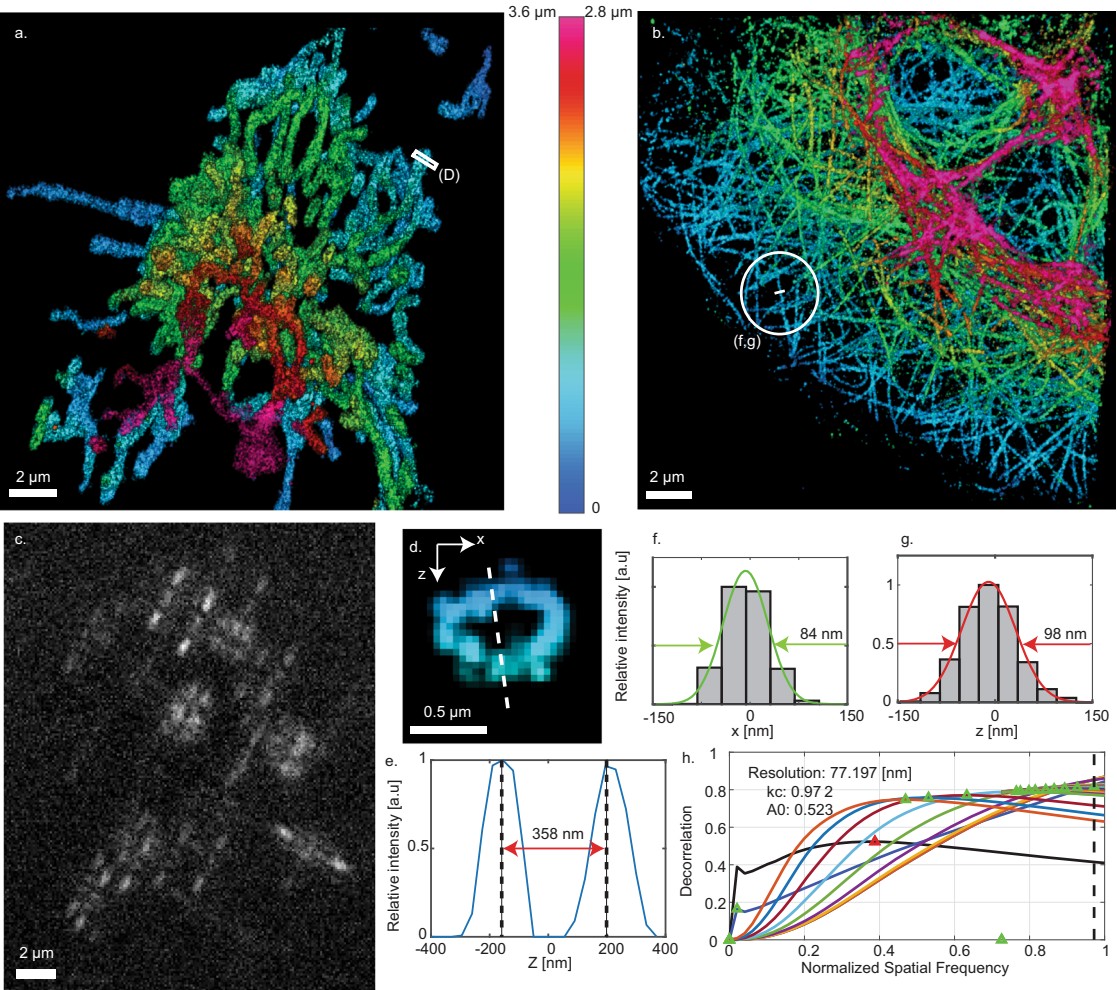

**Fig. 3 The results of the STORM experiments. a**, **b** Super-resolution reconstruction of mitochondria/microtubules in fixed U2Os cells. **c** An example frame from the experimental raw data; different Z-positions, manifested by different PSF shapes, can be observed. **d** An example of the hollow structure of the mitochondria. **e** The intensity histogram of the $X$, $Z$ hole-cross-section (shown in (**d**)). **f**, **g** The intensity histogram of one line from the microtubules reconstruction in the lateral and axial directions respectively. **h** Parameter-free image resolution estimation based on decorrelation analysis[28] yielded ~77 nm lateral resolution. 3D illustrations of the reconstructed mitochondria/microtubules are shown in Supplementary Movies 1 and 2.

serum (Biological industries), penicillin–streptomycin and glutamine at 37 °C and 5% $CO_2$. Cells were fixed with 4% paraformaldehyde and 0.2% glutaraldehyde in 1X PBS (37 °C, pH 7.3) for 45 min, washed, and incubated in 0.3 M glycine/PBS solution for 10 min. The cover glasses were transferred into a clean six-well plate and incubated in a blocking solution for 2 h (10% goat serum, 3% BSA, 2.2% glycine, and 0.1% Triton-X in 1X PBS, filtered with 0.45-μm PVDF filter unit, Millex). The cells were then immunostained overnight either with anti-TOMM20-AF647 (ab209606, Abcam) or with anti-α-tubulin-AF647 (ab190573, Abcam) and anti-β-tubulin-AF647 (ab204686, Abcam) diluted 1:250 in the blocking buffer. After staining, the samples were washed five times with PBS. To prevent detachment of the anti-tubulin antibodies, the sample was again treated with 4% paraformaldehyde and 0.2% glutaraldehyde in 1X PBS (pH 7.3) for 45 min, washed, and incubated in 0.3 M glycine/PBS solution for 10 min.

**STORM imaging and 3D super-resolution reconstruction**. For super-resolution imaging, a PDMS chamber was attached to a glass coverslip holding fluorescently labeled U2OS cells. Blinking buffer[33] 100 mM β-mercaptoethylamine hydrochloride, 20% sodium lactate, and 3% OxyFluor (Sigma Aldrich); in 1X PBS, adjusted to pH 8–8.5 was added and a clean coverslip was placed on top while minimizing any residual air bubbles in the chamber. In a typical experiment, we recorded 50,000 images with 30 ms or 50 ms exposure time to match the typical fluorescence on-time of the blinking emitters. The total duration of the measurement was 25 min and 40 min, respectively.

To localize the SMLM Tetrapod data, we used DeepSTORM3D, a neural net algorithm developed by our group[26]. Briefly, the net consisted of a fully convolutional model trained solely on simulated data. To ensure the imaging model closely matched the experimental data, we performed bead scans prior to each experiment to retrieve the aberrated pupil function using VIPR[32].

## Data availability
The data and software code supporting the findings of this study are available from the corresponding author upon reasonable request.

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

## Acknowledgements

We would like to thank D. Peselev, and O. Ternyak, and the Micro-Nano-Fabrication and Printing Unit at the Technion for assistance with the photolithography process, and XJet Ltd. for printing the ceramic mold used for the liquid immersed phase mask. We would also like to thank M. Bercovici and Y. Schechner for fruitful discussions. R.O.-K., E.N., B.F., O.A., and N.O. were supported by the European Research Council Horizon 2020 (802567). L.E.W. and Y.S. were supported by the Zuckerman Foundation; N.O. is also supported by the Israel Innovation Authority.

## Author contributions

R.O.-K., B.F., and Y.S. conceived the approach. R.O.-K. and Y.S. designed the liquid immersed phase mask, R.O.-K. fabricated the physical phase mask. R.O.-K., L.E.W., and Y.S. built the optical setup. R.O.K., O.A., L.E.W., and Y.S. collected the data. R.O.-K., E.N., and B.F. performed simulations and analyzed data. N.O. built the interferometry setup and performed interferometry measurements. R.O.-K., L.E.W., and Y.S. wrote the paper with contributions from all authors.

## Competing interests

The authors declare no competing interests.
