## [Peer Review File · Nature Communications]

REVIEWER COMMENTS

Reviewer #1 (Remarks to the Author):

The authors present a simple, versatile, and affordable method to produce high-quality optical phase masks. Considering that the phase delay of an optical wave-front depends just on the difference of optical path lengths, it becomes apparent that for achieving e.g. a smaller phase shift one can either reduce the physical length of the path, or reduce the index of refraction. By combining two materials with nearly matching refractive index, one can drastically reduce the phase shift for a given physical path length. This principle allows to increase the physical size of the features required to imprint a desired phase pattern on an optical wave-front. This offers new possibilities for building such phase masks, specifically one can use commercial 3D printing techniques to produce templates for such masks. The required precision constraints are lifted from the nanometer range to the micrometer range.

If one uses a liquid as index matching material, the door is open for adjusting the phase-mask to the needs of the user by just replacing the liquid. For example, one can scale the range of phase differences across the wave-front by adjusting the index of refraction. In the presented experiments, this approach is used to tune the working range of a tetrapod-point-spread-function (tetrapod-PSF). However, one could also think of tuning the range to a different wavelength range, by compensating the dispersion of the solid part of the mask. This issue is especially important for double-helix PSFs where sharp phase jumps are involved. These sudden phase jumps are strongly susceptible to dispersion and make mask production by conventional methods rather difficult and produce phase masks that are working only over a rather small spectral range. The presented new method would elegantly solve this problem.

The manuscript is written very clearly and concisely. The presentation of the concept and the experiments to validate the quality and performance of the printed liquid-immersion phase-masks are well-described and demonstrate the huge capabilities of this approach. However, I find it unfortunate, that the description of the mask production is only very briefly discussed in the manuscript (lines 73 to 77). Though it is presented in the supporting information, at least an overview of this process should be given in the Materials and Methods section.

Except for these minor issues I strongly recommend to accept the submission for publication.

Minor issues

Line 73: "... ceramic mold with resolution of few microns." Is this resolution isotropic or does it refer to the resolution in the height of the structure?

Line 160: "... 1 g l-1 D-Glucose" change to "1 g / l D-Glucose"

Line 178: "A typical experiment took 40\25 minutes, consisting of 50,000 images with 50\30 milliseconds exposure time."

Better write:

In a typical experiment we recorded 50,000 images with 30 ms or 50 ms exposure time. This took 25 min to 40 min, respectively.

Reviewer #2 (Remarks to the Author):

Orange et al present immersion diffractive optical elements (DOE), which allow less expensive fabrication of such devices. This is due to much larger length scales involved in generating the

required phase shifts, which is achieved by using immersion media on the diffractive optical element that is closely matched to the refractive index of the substrate. Examples of such DOEs are given for PSF engineering, which are leveraged for particle tracking and single molecule localization microscopy. I find this work to be of interest, it is a simple idea that could make a large impact. In essence, it can transform the length-scales from a diffractive optical element from the nanometer scale to micrometer, which eases fabrication. Further, tunability of the phase shift by exchanging the immersion media is shown, a capability which was before only reserved to spatial light-modulators. Some questions remain that should be addressed:

Does the PDMS phase mask, and its assembly with the fused silica wavers, need to be performed in a clean room? How do the authors deal with bubbles or voids, or other impurities?

While the authors state that the laser transmission is equivalent to conventional masks, I would assume that the various optical interfaces have each about $\sim 4\%$ reflection losses, unless they are coated. Were the silica wavers anti-reflection coated?

What is the optical flatness of the fused silica wavers and of the final sandwich structure (assembly of wavers and metal ring)? Would a small tilt affect wavefront quality, or just lead to some steering of the light?

While the authors show evidence of the proper working of their phase mask by showing the PSF, it would be important to measure the wavefront of the device. Could the authors perform a phase measurement, using either a Shack-Hartman wavefront sensor or off-axis holography? Such a measurement would address any doubts about the optical quality of their devices.

Lastly, are there concerns about the aspect ratio of the structures (i.e. if patterns of high aspect ratio could be faithfully manufactured and replicated with PDMS)? This could be an issue when manufacturing devices that have blazed or other grating structures with high line-spacings (i.e. Abrahamsson, Sara, et al. "Fast multicolor 3D imaging using aberration-corrected multifocus microscopy." *Nature methods* 10.1 (2013): 60-63.). Would the authors expect any issues in that regard?

Overall, enthusiasm for this manuscript is high. But more evidence of the wavefront quality and potential limitations for mask patterns would be welcomed.

Reviewer #3 (Remarks to the Author):

In the present manuscript, R. Orange et al. have experimentally implemented a tunable diffractive phase mask for super-resolution microscopy. Using the three-dimensional printing technique, they made a ceramic mold and formed the tetrapod and double-helix phase masks of transfer polymer. The developed phase mask seems to be an effective alternative to high-performance active phase control devices such as SLM and DMD. They demonstrated precision-tracking of a nano-particle and the super-resolution reconstruction of biomaterials. I expect the developed technique to be applied to a variety of studies thanks to the advancement of 3D printing and micro-fluidics. Thus, I believe this work could be suitable for *Nature Communications*, but I would like the authors to address the following minor concerns and comments:

1) As the authors mentioned, the effect of the fabrication imperfection on the accumulated phase error is similarly reduced as the refractive index difference decreases. However, the smaller the refractive index difference, the larger the height of the diffractive optical element. Then, with respect to equation

(2), the effect of the error in refractive index difference increases and highly precise control of the refractive index of the immersion liquid is necessary. Can the authors provide further discussion about this concern?

2) Following the above comment, what are the fabrication precision of the employed 3D printing technique and the precision of controlling the refractive index of the immersion liquid? Such information would make it possible to estimate the limits of the performance and precision of the developed diffractive phase mask.

3) In equation (2), Δh (Capital-delta-h) should be h.

4) The caption for Figure 3 should be improved. In particular, there is no explanation of the result of Figure 3F. Please provide more detailed and clear explanation/discussion of the results in Figure 3.

3D printable diffractive optical elements by liquid immersion

Author Response to Referees

We wish to thank the editor and reviewers for their interest and careful attention paid to our manuscript.

Reviewer comments are shown in blue italics; our response is in black. Extracts from the manuscript are in quotes with the key changes in red. These changes are highlighted in the “changes-marked” version of the revised manuscript as well.

Reviewer #1 (Remarks to the Author):

The authors present a simple, versatile, and affordable method to produce high-quality optical phase masks. Considering that the phase delay of an optical wave-front depends just on the difference of optical path lengths, it becomes apparent that for achieving e.g. a smaller phase shift one can either reduce the physical length of the path, or reduce the index of refraction. By combining two materials with nearly matching refractive index, one can drastically reduce the phase shift for a given physical path length. This principle allows to increase the physical size of the features required to imprint a desired phase pattern on an optical wave-front. This offers new possibilities for building such phase masks, specifically one can use commercial 3D printing techniques to produce templates for such masks. The required precision constraints are lifted from the nanometer range to the micrometer range.

If one uses a liquid as index matching material, the door is open for adjusting the phase-mask to the needs of the user by just replacing the liquid. For example, one can scale the range of phase differences across the wave-front by adjusting the index of refraction. In the presented experiments, this approach is used to tune the working range of a tetrapod-point-spread-function (tetrapod-PSF). However, one could also think of tuning the range to a different wavelength range, by compensating the dispersion of the solid part of the mask. This issue is especially important for double-helix PSFs where sharp phase jumps are involved. These sudden phase jumps are strongly susceptible to dispersion and make mask production by conventional methods rather difficult and produce phase masks that are working only over a rather small spectral range. The presented new method would elegantly solve this problem.

The manuscript is written very clearly and concisely. The presentation of the concept and the experiments to validate the quality and performance of the printed liquid-immersion phase-masks are well-described and demonstrate the huge capabilities of this approach.

1. However, I find it unfortunate, that the description of the mask production is only very briefly discussed in the manuscript (lines 73 to 77). Though it is presented in the supporting information, at least an overview of this process should be given in the Materials and Methods section.

We thank the reviewer for their enthusiasm regarding our method and appreciate the recommendation to provide more details in the materials and methods section. We have now added new sections on the fabrication and assembly process in the Materials and Methods section (shown below), and have expanded the Supplementary Information with step-by-step explanation about the fabrication process.

The changes in the Materials and Methods section:

3D printing and molding a phase mask:

A detailed description of the fabrication process is described in the Supplementary Information. Briefly, a ceramic mold of the reverse pattern of the phase mask was printed via additive manufacturing (Xjet

Ltd. Israel). A Polydimethylsiloxane (PDMS) cast was made and bonded to a high-quality fused silica optical glass (Siegert Wafer, Germany) via a brief plasma treatment to both the wafer and back of the PDMS mask.

To construct the liquid-immersion chamber, a PDMS frame was molded to a custom machined metal disk. To ensure a tight fitting with the inlet and outlet tubes, the frame was polymerized with a single tube inserted in the disk, which was cut after polymerization. The frame was then attached to both silica wafers, again by brief plasma treatments. This assembly is then used as the liquid-immersion chamber that contains the mask. The final step is to attach a light-blocking aperture to the back of the silica wafer in order to reject light beyond the size of the mask. This was done with UV-cured optical adhesive (NOA68T, Norland Products). The mask is then ready to be added to the optical system.

Adding and exchanging immersion liquid into the device:

To prepare the chamber for the injection of liquid, it was filled with isopropanol. This helps (1) reduce air bubbles and (2) remove any residual immersion liquid from previous uses. Next, the desired RI of the immersion liquid was prepared by mixing glycerol and water and injecting it into the chamber. At each step, the optical performance of the system was compared to the expected performance mainly by a comparison of the 3D-PSF range, i.e. the mask range is increased with a larger RI difference.

Except for these minor issues I strongly recommend to accept the submission for publication.

Minor issues

2. Line 73: "... ceramic mold with resolution of few microns." Is this resolution isotropic or does it refer to the resolution in the height of the structure?

We have now clarified the text to read "To fabricate the phase mask, we printed a ceramic mold with XYZ resolution of tens of microns (Supplementary Note IV)." While the manufacturer's specifications for the printing resolution is 50 μm axially and laterally, we don't expect the resolution to be isotropic. We have estimated that the axial precision, which is the most important dimension in this application, exceeded the specifications significantly, $\sim 15 \mu\text{m}$.

Fabrication precision:

When printing a template for phase mask fabrication, the key parameter is the relative error, i.e. the deviation from the upper profile of the printed part. While the precise error is difficult to quantify, the fabrication error for the instrument is quoted to be better than 50 μm in the axial and lateral directions. Nevertheless, we estimate the axial precision, which is the most important dimension in this application, to be closer to $\sim 15 \mu\text{m}$ or less. This is because, in practice, our mask's performance is similar to a photolithographically fabricated mask, and this is the axial precision required to obtain such results (see SI section IV). Notably, some fabrication error, specifically a vertical scaling in the mask, can be corrected for with the liquid immersion calibration process.

3. Line 160: "... 1 g l-1 D-Glucose" change to "1 g / l D-Glucose"

Fixed.

4. Line 178: "A typical experiment took 40\25 minutes, consisting of 50,000 images with 50\30 milliseconds exposure time."

Better write:

In a typical experiment we recorded 50,000 images with 30 ms or 50 ms exposure time. This took 25 min to 40 min, respectively.

Fixed.

Reviewer #2 (Remarks to the Author):

Orange et al present immersion diffractive optical elements (DOE), which allow less expensive fabrication of such devices. This is due to much larger length scales involved in generating the required phase shifts, which is achieved by using immersion media on the diffractive optical element that is closely matched to the refractive index of the substrate. Examples of such DOEs are given for PSF engineering, which are leveraged for particle tracking and single molecule localization microscopy.

I find this work to be of interest, it is a simple idea that could make a large impact. In essence, it can transform the length-scales from a diffractive optical element from the nanometer scale to micrometer, which eases fabrication. Further, tunability of the phase shift by exchanging the immersion media is shown, a capability which was before only reserved to spatial light-modulators.

Some questions remain that should be addressed:

1. Does the PDMS phase mask, and its assembly with the fused silica wafers, need to be performed in a clean room? How do the authors deal with bubbles or voids, or other impurities?

A key advantage of our approach is the relatively simplicity of assembly without the need for a clean room environment. We have now clarified the requirements in the text, including a new supplementary table with the specific pieces of equipment used. We have now described the steps in further detail including debugging steps, i.e. reducing bubbles (highlighted in yellow below). *This is mainly described in the supplementary Note II describing the fabrication of the mask:*

Note II. The Fabrication process of the immersion phase mask

The fabrication process can be done in standard conditions with the equipment listed in Supplementary Table 2, does not require a clean room, and is described below.

1. Fabricating a microscale mold: A ceramic mold (Zirconia) that contains the reverse pattern of the desired phase mask was printed via additive manufacturing (Carmel 1400, Xjet Ltd., Israel). An illustration is shown in Supplementary Fig. 2a.
2. Transferring the mold pattern to transparent material: The ceramic mold was first coated with a thin layer of oil (WD40) to reduce adhesion. Next, Polydimethylsiloxane (PDMS, SYLGARD 184, POLYMER G) was prepared by mixing the two-component elastomer and degassing in a vacuum chamber for ~30 minutes to remove bubbles. **The PDMS was then poured onto the lubricated mold, and placed into the vacuum chamber to facilitate additional bubble removal.** The PDMS was allowed to cure for 24 hours at room temperature on a leveled surface to ensure flatness. The polymerized PDMS was then carefully separated from the ceramic mold, and the outside edges were cleaned with a razorblade. An illustration is shown in Supplementary Fig. 2b.
3. Creating the PDMS frame: To create a mold for the outer chamber wall, we machined a metal disk with a slot to hold the liquid-exchange tubes. PDMS was poured on top of the mold holding the liquid exchange tube against a petri dish, and the assembly was debubbled, and cured as described previously. After 24 hours, the PDMS frame was separated from the metal disk with the tube embedded in the polymer. The tube was then cut at both inner edges of the wall, creating an inlet and outlet for the chamber. An illustration is shown in Supplementary Fig. 2c.
4. Assembling the liquid-immersed phase mask: A brief oxygen plasma treatment to PDMS and SiO₂ enables a sturdy, chemical attachment (~1 min per treatment, Zepto W6, Diener Electronic). This process was used to bond the phase mask and outer frame to one fused-silica wafer. Next, the chamber was closed by attaching the second wafer to the frame using the same process. Lastly, an M5 metal washer, with an inside diameter matching the phase mask diameter (5.3 mm), was glued to the external side of fused silica wafer. This reduced the effect

of light that might traverse the optical system at spatial frequencies beyond the desired range, e.g. supercritical-angle fluorescence. An illustration is shown in Supplementary Fig. 2d-f.

5. Adding immersion liquid to the chamber: The device is designed to contain a liquid with refractive index 1.4283 at 668 nm that must interface directly with the PDMS mask, index 1.4306 at 668 nm. **To minimize bubbles and voids created while filling the device, we first degas the solution in a vacuum chamber for 5 minutes and prefill the chamber with isopropanol.** The glycerol-water mixture is then injected into the chamber replacing the isopropanol. To compensate for any error in the RI of the PDMS or height scaling, the RI of the liquid was tuned iteratively, adjusting the ratio of glycerol and water and evaluating the optical performances of the system at each step by comparison to simulations of the desired 3D-PSF response.
6. Positioning the mask in the optical system: Our device is larger than a typical dielectric phase mask, but nonetheless requires a similar alignment procedure, namely placing the phase mask in the back focal plane. For fine alignment, we attached a threaded adapter to the assembly (SM1S10, Thorlabs) with optical adhesive (NOA68T, Norland Products) and mounted it on a 6-axis kinematic optic mount (Thorlabs K6XS) while monitoring the 3D response of the PSF.

Supplementary Table 2. **Instruments used in device fabrication and characterization**

Instrument	Model Number	Manufacturer	Purpose
Vacuum chamber	402020	Tarson	Degassing PDMS
Plasma cleaner	Zepto W6	Diener Electronic	Bonding PDMS to silica wafers
Refractometer	DR6200TF	A.Krüss Optronic	Characterizing immersion liquid

2. While the authors state that the laser transmission is equivalent to conventional masks, I would assume that the various optical interfaces have each about ~4% reflection losses, unless they are coated. Were the silica wafers anti-reflection coated?

The reviewer brings up an important question: does our device, which contains more interfaces, exhibit higher transmission losses due to additional reflections. This is now discussed in a new supplementary figure and explanation shown below. In brief, the effect of additional reflections is very small due to the relatively small differences in refractive indices inside the device. A note regarding the future improvement using antireflective coatings has been added to the text.

1. Signal transmission through the device interfaces:

The wafers in our liquid-immersion phase mask are high-quality fused silica, note that the same material was also used in our photolithography-based fabrication of phase masks. The transmission of the device is hampered mainly by reflections; however, since most reflections are between layers with very close refractive indices, the overall effect on transmittance is small, as detailed below.

According to Fresnel-coefficients equations, under the assumptions that (1) the incident angle of the wavefront is near 0, and (2) the transition between the PDMS to the liquid does add a reflection, we can calculate the power transmittance, T , following equations S3 and S4 noting that both polarization coefficients are equal due to assumption (1):

$$t = \frac{2n_1}{n_1+n_2}, \quad S3$$

$$T_{single\ polizarization} = \frac{1}{2} \frac{n_2}{n_1} |t|^2 \rightarrow T_{both\ polizarizations} = \frac{n_2}{n_1} |t|^2, \quad S4$$

where t is the amplitude transmission coefficient of the electric field. The mask structure is described in

Supplementary Figure 5. **simplified model of the liquid immersed chamber.** specifying the different layers and the corresponded refractive indices. T1, T2, T3, T4 are the power transmittance between the different interfaces.

Supplementary Fig. 5.

The total power transmittance is the product of all transition terms, T. For the liquid immersed phase mask, $T_{total} = T_1 \cdot T_2 \cdot T_3 \cdot T_4 = 0.9312$. For the photolithographically fabricated phase mask $T_{total} = T_1 \cdot T_4 = 0.9313$. Namely, the additional reflections at the interfaces lead to a signal reduction of 0.01%.

Notably, the most significant contributors to the total reflectance are the transitions with the largest refractive index changes, i.e. from air to fused silica ($T_1 = 0.9650$), and Fused Silica to air ($T_4 = 0.9650$), leading to a total reduction of ~7%. Of course, these transitions are also present for both phase mask types, but could be further reduced with antireflective coatings.

3. What is the optical flatness of the fused silica wavers and of the final sandwich structure (assembly of wavers and metal ring)? Would a small tilt affect wavefront quality, or just lead to some steering of the light?

The optical quality is now described in the SI in terms of three parameters provided by the manufacturer, the surface roughness (RA< 1 nm) the total thickness variation (TTV< 3.5 μm) and surface quality scratch-dig number (S/D of 20/10). Regarding tilt – we have experimentally measured a small tilt in the fabricated mask. Nevertheless, since no significant degradation in PSF quality (compared to simulation, or to a PSF obtained with a photolithographically fabricated mask) was observed, we conclude that the effect of this tilt is merely in slightly shifting the light laterally. We have added these parameters and measurements of the approximate tilt of our device to the SI.

1. Linear-phase estimation:

Although the assembly of the device was not done with specialized equipment, we did not detect significant aberrations in the PSF relative to the design. We used high quality fused silica wafers (surface roughness: $R_a < 1$ nm, surface quality: 20/10 scratch-dig number, total thickness variation < 3.5 μm), so the wafers should not add any unexpected aberrations. One potential assembly error of our bench-top device is in the non-parallelism in the chamber (illustrated in Supplementary Fig. 6), we measured the height differential across the device to be $< 0.2^\circ$, and $< 0.1^\circ$ across the printed-mask mold. The result of this slight height change across the device would manifest itself in a small linear phase accumulation that would shift the beam path.

Supplementary Figure 6. **An illustration of a possible deviation in parallelism of the different planes of the phase mask assembly (tilt is extremely exaggerated for clarity).** i the two silica wafers ii The PDMS frame iii The PDMS mask iv The liquid. The deviation in parallelism can causes from deviation in the printed mask which leads to deviation in the PDMS mask or from the PDMS frame polymerization.

4. While the authors show evidence of the proper working of their phase mask by showing the PSF, it would be important to measure the wavefront of the device. Could the authors perform a phase measurement, using either a Shack-Hartman wavefront sensor or off-axis holography? Such a measurement would address any doubts about the optical quality of their devices.

We very much appreciate this suggestion, and have added an off-axis holography measurement with an explanation to a new section and figure in the supplementary, showing good correspondence between the designed and fabricated phase masks.

1. Wavefront characterization by off-axis holography:

To measure the phase pattern produced by the liquid-immersed phase mask and compare it to the desired design, we used off-axis digital holography, which recovers the wavefront of a beam by backpropagation of a captured interference pattern. The optical setup is illustrated in Supplementary Fig. 4, a. The principle of off-axis holography is to capture an interference pattern rather than an image. This is done by splitting a collimated, coherent beam into two parts, namely an object beam and a reference beam, passing the object beam through the phase mask and then overlaying the beams at the sensor plane so that the interference pattern appears on the sensor. The captured interference intensity is described in Equation S2:

$$I(x, y) = |R|^2 + |O|^2 + (RO^* + OR^*) \quad \text{S2}$$

Where R, O stand for the complex wavefronts of the reference and object beams, respectively. $*$ denotes the complex conjugate operation. The term in the brackets is real-valued but contains the phase of the object beam, which is absent in a regular, intensity-only, imaging apparatus. The presence of the bracketed term provides information on the object beam wavefront at the sensor plane, which is enough to back-propagate it for reconstruction at any other plane. The mathematical process of reconstruction is described in detail by N. Verrier and M. Atlan¹.

Supplementary Figure 4. Wavefront characterization by off-axis holography. **a** Illustration of the off-axis digital holography setup for phase imaging. A laser beam is first spatially cleaned and collimated. The collimated beam is then split to an object and a reference beam. The object beam goes through the phase mask while the reference beam maintains its smooth wavefront, and both are recombined in an off-axis configuration to generate interference on the camera sensor. **b** The reconstructed wavefront of after the liquid immersed phase mask with the following parameters: $\Delta n=0.0018$, $\lambda = 514$. **c** The corresponding design of the liquid immersed phase mask. The color scale is in units of radians units.

5. Lastly, are there concerns about the aspect ratio of the structures (i.e. if patterns of high aspect ratio could be faithfully manufactured and replicated with PDMS)? This could be an issue when manufacturing devices that have blazed or other grating structures with high line-spacings (i.e. Abrahamsson, Sara, et al. "Fast multicolor 3D imaging using aberration-corrected multifocus microscopy." *Nature methods* 10.1 (2013): 60-63.). Would the authors expect any issues in that regard?

The reviewer brings up an important point, not all geometries can be manufactured well by additive manufacturing. That is, the aspect ratio of the printed structures is highly dependent on the printing method and the geometry of the printed part. To fabricate the desired mask well enough for sensitive applications, the parameters of the printing process need to be optimized. For multifocus imaging, a phase pattern such as the one shown in "Fast multicolor 3D imaging using aberration-corrected multifocus microscopy" is shown below.

To implement this geometry with the liquid immersed DOE using the printing process we employed, two heights are needed, axially separated by a 140 μm step, yielding a relative phase of π for $\lambda = 668 \text{ nm}$ and a refractive index difference of 0.0023. The most difficult geometry is likely to be the narrow white section (circled in red). If it is larger (laterally) than $\sim 40 \mu\text{m}$, this should work using our pipeline. Using the printing process employed here, the estimated aspect ratio is a height change of 150 microns over a lateral change of 40 μm . Therefore, assuming a back focal plane diameter of $\sim 5 \text{ mm}$, and judging by the figure from *Abrahamsson et al.*, implementing such a mask should be feasible or at least close to feasible. Notably, binary masks are less challenging to produce than a 3D structure that has many layers. We have now added a short discussion to the SI addressing this issue:

Note IV. Choosing a fabrication method

A brief explanation of the key considerations that factor into the choice of printing methods for a mask design are described below.

Phase mask suitability:

In this work, we used the Xjet printer, which has a maximum height change of 150 μm per 40 μm lateral change. This maximal aspect ratio proved to be sufficient for the DOEs we demonstrate, however there can be more challenging DOEs, e.g. optical gratings, where this could pose a limitation.

Overall, enthusiasm for this manuscript is high. But more evidence of the wavefront quality and potential limitations for mask patterns would be welcomed.

Reviewer #3 (Remarks to the Author):

In the present manuscript, R. Orange et al. have experimentally implemented a tunable diffractive phase mask for super-resolution microscopy. Using the three-dimensional printing technique, they made a ceramic mold and formed the tetrapod and double-helix phase masks of transfer polymer. The developed phase mask seems to be an effective alternative to high-performance active phase control devices such as SLM and DMD. They demonstrated precision-tracking of a nano-particle and the super-resolution reconstruction of biomaterials. I expect the developed technique to be applied to a variety of studies thanks to the advancement of 3D printing and micro-fluidics. Thus, I believe this work could be suitable for Nature Communications, but I would like the authors to address the following minor concerns and comments:

1) As the authors mentioned, the effect of the fabrication imperfection on the accumulated phase error is similarly reduced as the refractive index difference decreases. However, the smaller the refractive index difference, the larger the height of the diffractive optical element. Then, with respect to equation (2), the effect of the error in refractive index difference increases and highly precise control of the refractive index of the immersion liquid is necessary. Can the authors provide further discussion about this concern?

We thank the reviewer for this suggestion. We have now added additional discussion on the effect of scaling up the dimensions of the phase mask on the relative phase error to the supplementary:

The effect of scaling up mask dimensions:

As we reduce the refractive index difference, the heights of the mask should increase. Then the second term from equation S5 becomes more significant.

$$\delta\Delta\phi_{\text{Error}} \approx \frac{2\pi}{\lambda} (\Delta n \cdot \delta h + h \cdot \delta n) \quad \text{S5}$$

To estimate the tolerance in fabrication error at a specific refractive index difference between the mask and the surrounding media, we first calculate the $\delta\Delta\phi_{\text{Error}}$ of the photolithography phase mask, which we will use as a benchmark for acceptable phase error. The mean accumulated fabrication error measured in our photolithography fabrication was ~ 70 nm. With a $\Delta n \sim 0.5$, $\lambda = 668$ nm, we can calculate the relative phase error to be $\delta\Delta\phi_{\text{Error}} \approx \frac{2\pi \cdot 0.5 \cdot 0.07}{0.668} = 0.3292$ radians.

Now, we can estimate the fabrication precision of the current method required to achieve the same level of phase error. Here, $\Delta n = 0.0023$, $h_{\text{mean}} = 140$ μm . Assuming the precision of the glycerol-water concentration to be $\sim 1 \times 10^{-5}$, namely the precision of the refractometer, the fabrication error which would yield the same fabrication error as the photolithography case can be calculated to be:

$$0.3292 \text{ radians} \approx \frac{2\pi}{\lambda} (\Delta n \cdot \delta h + h \cdot \delta n) \quad \text{S6}$$

$$\delta h = \frac{0.5 \cdot 0.07 \mu\text{m} - 140 \mu\text{m} \cdot 0.00001}{0.0023} = 14.61 \mu\text{m} \quad \text{S7}$$

This parameter provides an estimation of the fabrication precision needed.

2) Following the above comment, what are the fabrication precision of the employed 3D printing technique and the precision of controlling the refractive index of the immersion liquid? Such information

would make it possible to estimate the limits of the performance and precision of the developed diffractive phase mask.

To address these issues we have added a section to the SI regarding the fabrication method which now contains three subsections discussing (1) the suitability of phase mask designs with respect to key parameters like high aspect ratios, (2) the fabrication precision, and (3) immersion-media characterization to the supplementary.

Phase mask suitability:

Does the aspect ratio of the DOE match the method of fabrication? In this work, we used the Xjet printer, which has a maximum height change of 150 μm per 40 μm lateral change. Our DOE design is well-suited for additive manufacturing in this respect as it is relatively smooth, but there are DOEs, e.g. optical gratings, where this will be a key limitation.

Fabrication precision:

When printing a template for phase mask fabrication, the key parameter is the relative error, i.e. the deviation from the upper profile of the printed part. While the precise error is difficult to quantify, the fabrication error for the instrument is quoted to be better than 50 μm in the axial and lateral directions. Nevertheless, we estimate the axial precision, which is the most important dimension in this application, to be closer to ~ 15 μm or less. This is because, in practice, our mask's performance is similar to a photolithographically fabricated mask, and this is the axial precision required to obtain such results (see SI section IV). Notably, some fabrication error, specifically a vertical scaling in the mask, can be corrected for with the liquid immersion calibration process.

Immersion-media characterization:

A concentration error of the immersion media results in a change in the refractive index. To monitor the mask-immersion liquid in the device, we characterized the refractive index with a commercial refractometer (DR6200TF, A. Krüss Optronic, GmbH), precision $\pm 1 \times 10^{-5}$. In practice, tuning the immersion media to a less precise extent $\pm 1 \times 10^{-4}$ was sufficient for our purposes.

3) In equation (2), Δh (Capital-delta-h) should be h.

Fixed, thank you.

4) The caption for Figure 3 should be improved. In particular, there is no explanation of the result of Figure 3F. Please provide more detailed and clear explanation/discussion of the results in Figure 3.

We expended the discussion of the results showing in manuscript figure:

“The range afforded by the Tetrapod mask was adjusted to enable the whole cell to be imaged simultaneously (Fig. 3 a,b), while still maintaining the precision to visualize the key features in the biological structures (Fig. 3 d,f,g). Fig. 3 d,e demonstrates qualitatively that our proposed design enables resolving mitochondrial hollow structures, in a similar fashion to results achieved with either a deformable mirror¹⁷ or a diffractive optical element²⁶ implementation. Furthermore, Fig. 3 f,g demonstrates a Gaussian fit to a single microtubule from our reconstruction which leads to a full width at half maximum (FWHM) of 84 nm laterally and 98 nm axially. The apparent width of a microtubule filament in Single molecule localization microscopy (SMLM) depends both on the localization precision and on the diameter of the fluorophore distribution around the microtubule center [cite paper below]. Modelling microtubules as cylindrical structures with a diameter of ~ 25 nm, labelled with fluorophores

attached via an antibody of size ~ 12.5 nm²⁷, indicate that our lateral and axial localization precisions are ~ 17 nm and ~ 24 nm respectively. This result is in line with previous work employing a deformable mirror implementation¹⁷. In addition, in Fig. 3 h we quantify the lateral resolution of the microtubules reconstruction to be ~ 77 nm using a parameter-free image resolution estimation algorithm based on decorrelation analysis²⁸.

We also have revised the figure as follows:

Figure 3. The results of the STORM experiments. **a & b** Super-resolution reconstruction of mitochondria/microtubules in fixed U2Os cells **c** An example frame from the experimental raw data; different Z-positions, manifested by different PSF shapes, can be observed. **d** An example of the hollow structure of the mitochondria. **e** The intensity histogram of the X, Z hole-cross-section (shown in d). **f & g** The intensity histogram of one line from the microtubules reconstruction in the lateral and axial directions respectively. **h** Parameter-free image resolution estimation based on decorrelation analysis²⁸ results in ~ 77 nm lateral resolution. 3D illustrations of the reconstructed mitochondria/microtubules are showing in Supplementary videos 1,3.

REVIEWERS' COMMENTS

Reviewer #1 (Remarks to the Author):

The authors have perfectly addressed and answered all my remarks and questions. I recommend publication as is - it is an excellent manuscript about an important topic.

Reviewer #2 (Remarks to the Author):

The authors have addressed my concerns thoughtfully and thoroughly. Some questions and suggestions for the supplementary information have come up:

Supplementary Figure 2, caption: I find the notation b (c) confusing. I would recommend to split it up into b and c. If necessary, more letters could be used to label the sub-panels of b and c for clarity.

Suppl. Figure 3: I got at first confused with the figure caption using the notation a(b). I would recommend to separate the labels a and b in the caption. Secondly, I assume figure panel b shows the results with the liquid immersed phase mask. Qualitatively, the image looks somewhat distorted compared to the result with the photolithographically manufactured mask. Does the defocus accentuate small wavefront errors? It was also not clear why the defocus was added. Did the authors want a more spread out intensity pattern for their measurements?

For supplementary Figure 4: could the slightly different appearance (i.e. saddle, peaks/valleys) between figure 4b and c result from different offsets in phase? For me it is hard to judge if the two are the same or show significant differences, but this could also be due to the wrapping of the color map and a global phase offset.

Alternatively, a surface plot or line profiles could be added.

Otherwise I am very happy with the manuscript and look forward to its publication.

Sincerely,
Reto Fiolka

Reviewer #3 (Remarks to the Author):

I have carefully read the revised paper and supplementary information. The author has adequately addressed all the questions and concerns that I had raised in the previous report. I would be happy to recommend the publication of this paper by R. Orange-Kedem in the current form.

3D printable diffractive optical elements by liquid immersion

Author Response to Referees

We want to thank again to the editor and reviewers for their comments and interest in our manuscript.

Reviewer comments are shown in blue italics; our response is in black. Extracts from the manuscript are in quotes with the key changes in red. These changes are highlighted in the “changes-marked” version of the revised manuscript as well.

Reviewer #1 (Remarks to the Author):

The authors have perfectly addressed and answered all my remarks and questions. I recommend publication as is - it is an excellent manuscript about an important topic

We want to thank the reviewer, we appreciate their recommendation.

Reviewer #2 (Remarks to the Author):

The authors have addressed my concerns thoughtfully and thoroughly.

Thanks.

Some questions and suggestions for the supplementary information have come up:

Supplementary Figure 2, caption: I find the notation $b(c)$ confusing. I would recommend to split it up into b and c . If necessary, more letters could be used to label the sub-panels of b and c for clarity.

Fixed.

Suppl. Figure 3: I got at first confused with the figure caption using the notation $a(b)$. I would recommend to separate the labels a and b in the caption.

Fixed.

Secondly, I assume figure panel b shows the results with the liquid immersed phase mask. Qualitatively, the image looks somewhat distorted compared to the result with the photolithographically manufactured mask. Does the defocus accentuate small wavefront errors? It was also not clear why the defocus was added. Did the authors want a more spread out intensity pattern for their measurements?

We thank the reviewer for this question. Given the excellent performance of the liquid-immersed PSFs measured using fluorescent beads in the microscope (e.g. main text fig.1, it is likely that the distortion observed in this figure's measurement is due to imperfection in the setup used here, or in its alignment. We agree that it is possible that the thickness of the mask accentuates these wavefront aberrations. Please keep in mind that the purpose of this specific experiment was to quantify photon efficiency; for quantifying other aspects of the DOE performance there are additional experiments (main text figures 1 and 3, SI figure 4).

For supplementary Figure 4: could the slightly different appearance (i.e. saddle, peaks/valleys) between figure 4b and c result from different offsets in phase? For me it is hard to judge if the two are the same or show significant differences, but this could also be due to the wrapping of the color map and a global phase offset.

Alternatively, a surface plot or line profiles could be added.

Otherwise I am very happy with the manuscript and look forward to its publication.

We added more panels to supplementary figure 4, presenting different profile lines of both the model and the reconstruction images. It is important to mention that the mismatches between the model and the reconstruction can occur from at least 3 factors: 1. A physical effect in the mask, caused by one or more of the following: printing error of the template, converting the template to PDMS mask, errors in the concentration of the liquid, bubbles and more. 2. The alignment of the interferometry optical system. 3. The reconstruction algorithm (for example the high frequency lines on the reconstructed mask).

As we can see from the center line profiles the peak in the center of the mask is lower in the reconstructed liquid-immersed mask, however the side lines profiles are highly matched. This might be caused by deviations from planarity of the illuminating wave in the interferometry setup.

Revised Supplementary Figure 4:

Supplementary Figure 4. **Wavefront characterization by off-axis holography.** **a** Illustration of the off-axis digital holography setup for phase imaging. A laser beam is first spatially cleaned and collimated. The collimated beam is then split to an object and a reference beam. The object beam goes through the phase mask while the reference beam maintains its smooth wavefront, and both are recombined in an off-axis configuration to generate interference on the camera sensor. **b** The reconstructed wavefront of the liquid immersed phase mask with the following parameters: $\Delta n=0.0018$, $\lambda = 514$. **c** The corresponding design of the liquid immersed phase mask. Profile plot of the relative phase of: **d** Side line of the reconstructed mask presented in **b**. **e** Center line of the reconstructed mask presented in **b**. **f** Side line of the design presented in **c**. **g** Center line of the design presented in **c**.

Reviewer #3 (Remarks to the Author):

I have carefully read the revised paper and supplementary information. The author has adequately addressed all the questions and concerns that I had raised in the previous report. I would be happy to recommend the publication of this paper by R. Orange-Kedem in the current form.

We thank the reviewer, and we appreciate their suggestions and the final recommendation.